# Effects of Teaching Games on Decision Making and Skill Execution: A Systematic Review and Meta-Analysis

**DOI:** 10.3390/ijerph17020505

**Published:** 2020-01-13

**Authors:** Manuel Tomás Abad Robles, Daniel Collado-Mateo, Carlos Fernández-Espínola, Estefanía Castillo Viera, Francisco Javier Giménez Fuentes-Guerra

**Affiliations:** 1Faculty of Education, Psychology and Sport Sciences, University of Huelva, 21071 Huelva, Spain; manuel.abad@dempc.uhu.es (M.T.A.R.); carlos.fernandez@ddi.uhu.es (C.F.-E.); estefania.castillo@dempc.uhu.es (E.C.V.); 2Centre for Sport Studies, Rey Juan Carlos University, 28943 Madrid, Spain; danicolladom@gmail.com

**Keywords:** TGfU, sport pedagogy, technique-focused approaches, tactical approaches

## Abstract

The question of how games should be taught is still a controversial subject. There has been a growing number of studies on teaching games and coaching sports since the first publication of Bunker and Thorpe on Teaching Games for Understanding (TGfU). In this sense, the present systematic review and meta-analysis aimed to systematically review the scientific literature about the effects of technical and tactical approach interventions on skill execution and decision making, and to examine the influence of the teacher/coach management style. A systematic literature search was carried out in accordance with PRISMA guidelines in Web of Science (WOS), PubMed (Medline), Scopus, and SportDiscus electronic databases. A total of seven and six studies were deemed to meet the inclusion criteria for decision making and skill execution, respectively. Meta-analysis results showed that tactical interventions achieved significant improvements in decision making (effect size = 0.89 with 95% confidence interval (CI) from 0.12 to 1.65), but they did not show significant improvements in skill execution (effect size = 0.89 with 95% CI from −0.45 to 2.23) compared to technical approaches. However, the heterogeneity of interventions was large and the quality of evidence was low according to GRADE. In conclusion, tactical approaches are recommended to teach games and sports in order to develop technique, understanding, tactical knowledge, and decision making, which are demanded in game play. These findings could be useful for teachers and coaches to improve these aspects of their players and students.

## 1. Introduction

The best way to teach games is still a controversial subject around the world [1]. Since the first publication of Bunker and Thorpe on Teaching Games for Understanding (TGfU) [2], many studies have been conducted to evaluate the effects of different types of models. Therefore, multiple approaches based on game teaching and coaching have emerged as an alternative to technique-focused approaches aimed at solving the potential problems related to the development of technique at the expense of tactical knowledge and decision making [3]. In technique-focused, traditional, or skill-based approaches, technical skill is pre-determined and based on a perfect model of execution where players execute the skill in a repeatable manner [4] in isolation from the game context and are trained until it is performed well enough to play the game [5]. Moreover, technique-based approaches “focus first on the teaching of the techniques of the game before going on introducing tactical knowledge, once a skilled background has been developed” [6] (p. 40).

This has led to the use of broad terms in tactical approaches or game-based approaches, which, in spite of some small differences, share some common main ideas (they focus on the game as a whole, where they place learning in modified games, and there is an emphasis on questioning to stimulate thinking and interaction) [7]. Some of the better known game-based approaches that follow TGfU are the Tactical Game Approach, Game Sense, Play Practice, Games Concept Approach, Tactical-Decision Learning Model, ball School model, and Invasion Games Competence Model. For these approaches, “the main element is that key learning occurs from the game itself and game-related activities, as opposed to drills completed in isolation then applied during a game” [8] (p. 65). However, tactical approaches underline the complementarity of technical and tactical dimensions of skilled performance [9] and they aim to not only teach the skills required to play a game, but also to allow one to develop the ability to understand the game’s tactics and strategies [1]. As stated above, central to the criticisms of the technique-based model was the development of inflexible techniques that did not enable the student or player to resolve real game situations. Therefore, there may be a lack of transfer from practice to games [10]. On the other hand, in tactical approaches, “skill execution is not neglected but developed after understanding the game’s strategies and tactics” [11] (p. 30). In this way, these approaches have been related to constructivist and situated learning theories [12], where the student’s knowledge construction takes place in games, solving problems, and reflection [13]. Therefore, tactical approaches focus on student learning within a game context and permit people to develop a tactical understanding of the game, tactical awareness, decision making, and skill execution [13].

The research focused on comparing technique-focused and game-based approaches to teach games has increased in recent decades around the world [6,14]. In this way, there has been considerable discussion and research on the most effective method to teach games, and many studies have focused on comparisons of tactical and technical approaches [11].

In the scientific literature, the effects of both types of interventions on several variables have been studied, mainly skill execution and decision making, comparing the two pedagogical approaches (tactical and technical), in order to identify which one can achieve greater results [15]. In this regard, Rovegno et al. [16] highlighted the relationship between motor skill execution and decision making. Nevertheless, previous studies have provided controversial results on the development of skill execution and decision making when technical and tactical models of teaching games are used. Therefore, the up-to-date scientific literature does not provide clear guidelines about the most adequate or optimal approach. In this sense, the comparative approach has much to offer, and it seems clear that there is still a need to identify effective ways to teach students and players in order to develop both game play and participation [11].

To provide clarity on the topic, Oslin and Mitchell [17] published a review of studies evaluating game-centered approaches to teaching and coaching. They highlighted several core concepts to justify the use of this model, including the development of decision making skills and effective decision makers. The central findings section of their review provides an in-depth discussion of the studies comparing technical and tactical approaches. On the other hand, Harvey and Jarrett [14] published a review where they noted that several key challenges remain within game-centered approach research (in-depth inquiry on tactical approaches in coaching contexts, further assessment of tactical awareness development, and the use of longitudinal research designs, among others). These previous reviews have emphasized that results concerning the development of skill execution with tactical and technical approaches are equivocal. In this sense, it is important to emphasize that, in the implementation of technical and tactical approaches, the role of the teacher/coach is very relevant, given that physical education teachers and coaches do not present the same management style (e.g., teacher/coach’s personality, communication skills, use of feedback, motivation, etc.). In this sense, there is a lack of information about coaches’ behaviors in teaching games.

Therefore, given the interest and relevance of the topic, as well as the controversy about the best way to teach games and the importance of the teacher/coach behavior style, a systematic review and meta-analysis is necessary. Based on previous reviews [6,14,15,17], it can be hypothesized that a tactical approach will result in better learning outcomes than a technical approach in teaching games. However, the magnitude of those differences must be quantified and proper analyses must be conducted to accept or reject that hypothesis. To the best of our knowledge, no meta-analysis of studies comparing skills-based and tactics-based approaches to teaching games has been performed before the current one. In this regard, the purpose of the present systematic review and meta-analysis was to systematically review the scientific literature about the effects of technical and tactical approach interventions on skill execution and decision making, and to examine the influence of the teacher/coach management style.

## 2. Methods

The Preferred Reporting Items for Systematic Review and Meta-Analysis Protocols (PRISMA) guidelines have been followed to conduct the current review [18].

### 2.1. Inclusion Criteria

The manuscripts were deemed eligible for inclusion if they met the following criteria: (1) the intervention was based on a comparison of the technical and tactical models in sports education; (2) players or students’ decision making and/or skill execution were measured; (3) articles were written in English or Spanish; (4) manuscripts were published in the XXI century; and (5) articles were original research (not a systematic review or literature analysis). To reduce selection bias, each study was independently reviewed by two of the authors (C.F.-E. and M.T.A.R.), who mutually determined whether or not they met basic inclusion criteria. If a consensus could not be reached on inclusion of a study, the matter was settled by consultation with a third author (F.J.G.F.-G.).

### 2.2. Search Strategy

A systematic literature search was carried out in accordance with PRISMA guidelines [18] in Web of Science (WOS), PubMed (Medline), Scopus, and SportDiscus electronic databases. The search was conducted from the year 2000 to May 2019. The following syntax was used for the search process: (“TGFU” OR “teaching games for understanding” OR “tactical games approach” OR “tactical approach” OR “tactical games model” OR “game centred approach” OR “game sense approach” OR “game based approach” OR “games teaching” OR “constructivis*”) AND (“sport” OR “physical education” OR “training”) AND (“techniques” OR “technical skills” OR “traditional Model” OR “technical approach” OR “skill-centred approach” OR “instructional model*” OR “instructional method*” OR “instructional coaching”) AND (“decision making”) AND (“intervention” OR “experimental” Or “quasi-experimental” OR “randomized controlled trial”).

### 2.3. Assessment of Risk of Bias

To evaluate the risk of bias, the PEDro scale [19] was used. This scale was developed to assess the quality of intervention studies, especially randomized controlled trials. The GRADE approach, which involves a four-point scale (“very low”, “low”, “moderate”, and “high”), was used to assess the quality of evidence [20]. In this approach, the quality of the evidence is downgraded when inconsistency, indirectness, imprecision, or publication bias are present. Table 1 shows the risk of bias results of included articles. To evaluate the risk of bias and the quality of evidence, each study was independently reviewed by two of the authors (C.F.-E. and M.T.A.R.). If a consensus could not be reached, the matter was settled by consultation with a third author (F.J.G.F.-G.).

### 2.4. Data Collection

Firstly, two authors extracted data from the included articles. Subsequently, the gathered information was checked by another author. Following the recommendations from PRISMA guidelines, relevant information included participants, intervention, comparisons, results, and study design (PICOS) [21]. Table 2 shows the main characteristics of the different protocols of intervention and the essential participants’ characteristics (sex, age, sample size, level of education or setting, and treatment).

Concerning interventions, we summarized the following details: duration, number of lessons, type of intervention program (tactical or technical), and teacher/coach management style analysis. In the study by Guijarro-Romero et al. [22], students received a 5 weeks of learning program, consisting of 10 lessons and 1 h per lesson. Students consecutively performed two teaching units (indoor football and basketball) based on a traditional technical-tactical approach. The tasks administered were more instructive and focused on learning isolated and without connection from one sport to another. Students received fully instructive feedback during the different sessions. With regard to the tactical approach, students carried out an intervention program using alternate teaching units of indoor football and basketball based on a tactical approach. The intervention consisted of carrying out a session of indoor football followed by another of basketball, focusing on establishing a connection in the learning of both sports. The teacher/coach management style was not analyzed.

In the study by Ashraf [23], students received a 2 months of learning program, but the number of lessons was not reported. The technical approach used was the traditional method, and Teaching Games for Understanding was used, although the authors do not describe its characteristics. The teacher/coach management style was not analyzed. On the other hand, the Morales-Belando and Arias-Estero’s study [24] lasted 2 weeks, with 11 lessons of 80 min. “Technical group’s lessons followed the traditional segments in sailing: (a) The coach taught the knowledge out of the water; (b) the sailors then applied such knowledge to a situation in which they sought to improve skill execution; and (c) finally, the participants practiced in a race” [24] (p. 4). The traditional approach only focused on how to act, so the technical content was taught first and the tasks were stripped of a real race context and the coach told the sailors what they must do. Moreover, the learners had a passive role, carried out the coach’s orders, and tried to imitate a perfect technical approach, whereas the coach told them and showed them what and how to do the tasks, while using technically perfect actions [24]. On the other hand, “the TRfU group’s lessons were created following the lesson segments: (a) The teacher set up the “race form” so that participants would work on the tactical aspect similar to a real race; (b) the teacher conducted “teaching for understanding” so that the children could reflect on what they had to do and why; (c) the teacher conducted “drills for skill” development so that participants could improve their skill execution; (d) the class returned to the “race form” so that the participants could perform a lesson segment very similar to the initial lesson segment; and (e) the teacher conducted a “review and closure” so that participants could reflect on the integration and understanding of skill execution and decision making” [24] (p. 4). The teacher/coach management style was not analyzed.

In addition, in Nathan’s research [25], the students received two lessons per week comprising 40 min per lesson for 5 weeks. In this study, Skill Technical was used, which is a teacher-centered approach based on the practice of skill drill activities of movement skills in an isolated way. This conceptual framework emphasizes the importance of teaching and learning skills prior to game play through skill drill practice [26]. On the other hand, Teaching Games for Understanding was used: Including the performer, environments, and task, where decision making and skill execution are derived from the game concept and thinking strategically. Moreover, Nathan [25] investigated the teachers’ reflections and experiences about questioning. In relation to Gray and Sproule’s study [27], this research lasted 5 weeks, with five lessons and 60 min per lesson. In this study, the skill-focused approach followed the physical education department’s program for teaching basketball. The teacher’s overall aim was to develop the pupils’ performance in 4v4 games following “his own knowledge and beliefs about teaching to deliver the program set out by the physical education department” [27] (p. 19). The tactical approach used “emphasizes tactical understanding and the development of motor skills as a means of solving tactical problems within a game-practice-game format. The teacher decides on the tactical problem that has to be addressed and presents games and practices that both emphasize the specific tactical problem” [27] (p. 19). The teacher/coach management style was not analyzed.

On the other hand, in the study by Psotta and Martin [28], the students received a 5 weeks of learning program, consisting of 10 lessons and 90 min per lesson. In this study, a technical-tactical model with an emphasis on orientation to tactical skills was used: “the technical skills are taught under controlled conditions in a predictable learning environment, and the tactical skills are taught using an unpredictable environment” [28] (p. 8). On the contrary, in the technical-tactical model with an emphasis on orientation to technical skills, the technical skills are taught in an unpredictable environment, and the tactical skills are taught using a match, with a teacher’s verbal instructions being related to tactics [28]. The teacher/coach management style was not analyzed. Finally, the research by Chatzopoulos et al. [29] lasted 5 weeks, and consisted of 15 lessons and 45 min per lesson. In this study, the technique group began the lesson with a demonstration of a specific technique, followed by practice of the technique in a series of drills. Following this, a tactic was taught for 5 min. The teacher introduced a tactical element on a blackboard and then on the game field. On the other hand, the games group began with a modified game designed to stimulate tactical thinking [29]. Next, technique (through drills) and tactical instruction were allocated. The teacher/coach management style was not analyzed.

### 2.5. Statistical Analysis

In these meta-analyses, a random-effects model was used to measure the effect of interventions based on technical and tactical approaches on decision making and skill execution. Figures 2 and 3 show the results of each study on these variables. The effect size was calculated using means and standard deviations before and after treatment [30]. For these meta-analyses, the magnitude of Cohen′s d was specified as follows: (a) “large”, for values greater than 0.8; (b) “moderate”, when it was between 0.5 and 0.8; (c) and “small”, for values between 0 and 0.5. Heterogeneity was evaluated by calculating the following statistics: (a) *p*-value of Cochran′s *Q*-test and (b) *I*^2^, which is a transformation of the H statistic used to determine the percentage of variation which is caused by heterogeneity. The most common classification of *I*^2^ considers values higher than 50% as large heterogeneity, values between 25% and 50% as average, and lower than 25% as small [31]. The tool Review Manager 5.3 was used to conduct all analyses [32].

## 3. Results

### 3.1. Study Selection

Figure 1 (PRISMA flow diagram) shows the complete process followed in the current systematic review. The original search identified a total of 51 manuscripts from the electronic databases: WOS (11), PubMed (4), Scopus (28), SportDiscus (7), and additional records identified through other sources (1). Five of them were removed because they were duplicated. Subsequently, to find any additional articles that met the inclusion criteria [33], the reference lists of articles retrieved and other sources were screened as part of a complementary search. One additional manuscript was found. Of the remaining 46 articles, 30 were removed because they were not connected with the study theme, five because the intervention programs did not have a technical group or only an experimental group, one because it did not conduct baseline evaluations, and three because they were systematic reviews or literature analyses. Therefore, the final number of studies included for meta-analyses was seven and six for decision making and skill execution, respectively (Figure 1).

### 3.2. Risk of Bias

Table 1 shows the risk of bias of the six selected articles according to the PEDro scale. Scores varied from six to seven [22,27,28] and to eight [23,24,25,29]. Regarding the quality of evidence, the GRADE guidelines have been followed. In this sense, the quality of evidence was downgraded twice: firstly, due to the high degree of heterogeneity, and secondly, because of the relatively low number of participants in the studies. However, it was upgraded because the total effect size was 1.78 and 1.86 for decision making and skill execution, respectively. Therefore, the quality of evidence according to the GRADE guidelines was “low”, which was defined as “Our confidence in the effect estimate is limited: The true effect may be substantially different from the estimate of the effect” [20] (p.404).

### 3.3. Study Characteristics

Table 2 shows a summary of the characteristics of the study. There was a total of 357 participants. Of these, 180 were distributed in the technical group (TEG) and 177 were allocated the tactical group (TAG). Two studies were conducted in primary school, one in middle school, one in secondary school, two in a university, one in a sailing school, and one in a badminton school.

### 3.4. Interventions

The tactical approaches used were the tactical approach, TGfU, Teaching Races for Understanding (TRfU), TGfU revised, game-based approach, and technical-tactical model with an emphasis on orientation to tactical skills. Regarding the technical models, these were as follows: Technical approach, traditional method, traditional teaching mode, Skill Drill Technical, skill-focused approach, and technical-tactical model with an emphasis on orientation to technical skills.

The intervention duration varied between 2 and 5 weeks. The number of lessons ranged between 5 and 15. The study conducted by Ashraf [23] did not specify the number of lessons, and only specified the intervention duration (2 months). Only one study [25] investigated the teachers’ reflections on and experiences of questioning.

### 3.5. Outcome Measures

Figure 2 and Figure 3 show the effects of technical and tactical approach interventions on participants’ skill execution and decision making. To evaluate the skill execution and decision making, two articles used the Game Performance Assessment Instrument (GPAI), and another developed an adapted instrument from this: the Race Performance Assessment Instrument (RPAI). Another manuscript used the Game Performance Evaluation Tool (G-PET), only to assess decision making; another used an adaptation of the Game Play Observational Instrument (GPOI); another used a coding instrument (CI); and another used the Soccer Performance Observation System (SPOS) based on GPAI.

As can be seen in Figure 2, the meta-analysis results showed that tactical intervention resulted in improvement compared with the comparison groups in almost every article (see Figure 2) and was significant in five studies [22,23,24,27,29]. However, the study by Nathan [25] reported no significant difference between TEG and TAG models in terms of skill execution or decision making. The overall effect size for decision making was 0.89, with a 95% CI from 0.12 to 1.65. Following the proposed classification, this effect size was large. The heterogeneity level was large (*I*^2^ = 99%) and the *P*-value of the Cochran *Q*-test was <0.01.

Figure 3 shows that two of the six articles reported significant improvements in skill execution relative to the baseline, caused by the tactical intervention [24,27]. One article was removed because it did not evaluate the skill execution [23]. Moreover, no studies reported significant improvements in skill execution because of technical treatment. The overall effect size for skill execution was 0.89, with a 95% CI from −0.45 to 2.23 (see Figure 3). In accordance with the proposed classification, this effect size was large. The heterogeneity level was large (*I*^2^ = 100%) and the *p*-value of the Cochran *Q*-test was <0.01. The effects of tactical or technical treatment on participants’ decision making and skill execution are shown in Figure 2 and Figure 3.

## 4. Discussion

The present systematic review and meta-analysis aimed to systematically review the scientific literature about the effects of technical and tactical approach interventions on skill execution and decision making, and to examine the study of the influence of the teacher/coach management style. The first main result showed that the tactical approach resulted in a significant improvement in decision making compared to the technical approach. This significant enhancement was observed in five of the seven analyzed studies and can be considered as large, according to the overall effect size (total effect size of 0.89, with a 95% CI from 0.12 to 1.65 and *p*-value = 0.02). In this regard, researchers consider that a tactical approach can make a significant contribution to the development of several areas of play such as tactical understanding and decision making [34]. However, there is still limited research about whether coaches are aware of the methodologies which may improve tactical understanding and decision making [35].

Another main finding was that the tactical interventions also achieved significant benefits in skill execution. In this regard, the overall effect size for skill execution was 0.89, with a 95% CI from—0.45 to 2.23 and *p*-value = 0.19. The technical approach did not result in a significant improvement in skill execution in any studies evaluated, but the tactical model resulted in a significant improvement in skill execution in two of six studies. In this respect, Bunker and Thorpe [2] proposed the integration of skills into contextualized situations in an attempt to link tactics and skills within a game context. Therefore, the tactical approach has the potential to facilitate the development of technical skills and tactical knowledge [36]. Therefore, this model may be more adequate to improve not only decision making, but also the skill execution, compared to the technical approach. Accordingly, findings from the current meta-analysis could be useful for teachers and coaches to improve skill execution and decision making among players and students. Nevertheless, due to the large heterogeneity and the low quality of the evidence, interpretation of this meta-analysis must be conducted with caution. In this sense, “more work needs to be undertaken to reinforce and further demonstrate the relationship between game centred training and skill development” [8] (p. 68).

One of the included articles showed very outstanding results in favor of the tactical group in decision making using TGfU [23]. In addition, another study [27] showed outstanding results in skill execution favor of the tactical group. This research, with five lessons of 60 min of duration during five weeks, focused on “tactical understanding and the development of motor skills as a means of solving tactical problems within a game-practice-game format” [27] (p. 19). Given the great effect observed in these studies, future research may focus on corroboration of the benefits of these protocols. In this sense, according to Forrest [13], the actual meaning of employing tactical approaches (and technical approaches) has been little explored and more research is needed to clarify what we really do when we implement these models.

Regarding the characteristics of the successful tactical interventions, the duration of the interventions varied between 2 and 5 weeks, between 5 and 15 lessons, and between 40 and 90 min per lesson. Therefore, according to these results, the benefits of tactical interventions could not be linked to the treatment length. With regards to this point, it is important to note that the implementation of technical and tactical approaches can be problematic, given that it depends on the teacher who is in charge, rather than just the model used [13]. This is more complicated when teachers and coaches think they use an alternative approach and are actually using a traditional method [37].

As can be seen in Table 2 and Figure 2, significant improvements in decision making from tactical models are not related to participants’ age or education level/setting. On the other hand, the results showed significant improvements in skill execution using tactical models among secondary students and school sports players (see Table 2 and Figure 3). Therefore, these results showed that tactical approaches can be used to improve skill execution. Nonetheless, it must be noted that the development of tactical and technical approaches is profoundly related to the environment and the fact that each context is different [7]. In this sense, physical education teachers and coaches do not present the same pedagogical characteristics (e.g., use of knowledge in practice and differences in training times) [38]. Moreover, transitioning to tactical approach pedagogy is challenging and can lead to frustration [38].

Concerning the type of sport, there were significant improvements in decision making using tactical approaches in all of them, except badminton [25] and soccer [28]. Regarding skill execution, there were significant improvements in the tactical group for basketball [27] and sailing [24]. It is important to note that the study focused on sailing that achieved significant between-group improvements in decision making and execution in favor of the tactical approach.

To our knowledge, this is the first systematic review and meta-analysis aimed at comparing the effects of technical and tactical approaches in decision making and skill execution, using a strong and widely-accepted methodology (PRISMA) and providing conclusions based on the existing evidence. Although relevant results were observed and a tactical approach can be strongly recommended based on the findings, further studies are needed to increase the quality of the evidence and to clarify what teachers and coaches do when they implement these models. In this sense, in relation to analysis of the influence of the teacher/coach management style, only one study [25] investigated the teachers’ reflections on and experiences of questioning. Future research could also examine the teaching and learning processes involved when adopting different approaches to teaching in order to know the effect that teachers’ personalities may have on students’ learning [29].

Nevertheless, the present systematic review with a meta-analysis has some limitations. First, four studies used the same instrument (GPAI) or an adapted version of it, but the other two–three studies were carried out with other instruments to assess skill execution and decision making (respectively). Second, the literature search was limited to two languages: Spanish and English. Therefore, the risk of the exclusion of manuscripts written in other languages was high. Finally, the meta-analysis showed a high level of heterogeneity, which means that the interpretation of the results of this study must be considered with caution.

## 5. Conclusions

Tactical approaches can be strongly recommended to teach games and sports in order to better improve skill execution and decision making, which are demanded in game play. In this regard, tactical approach interventions are useful for improving the players’ and students’ decision making, while technical models may be inadequate. On the other hand, tactical models could have positive effects on skill execution. Nevertheless, there is a lack of information about teacher/coach management style. These findings could be useful for teachers and coaches, but must be considered with caution given the heterogeneity and the low quality of the evidence.

## Figures and Tables

**Figure 1 ijerph-17-00505-f001:**
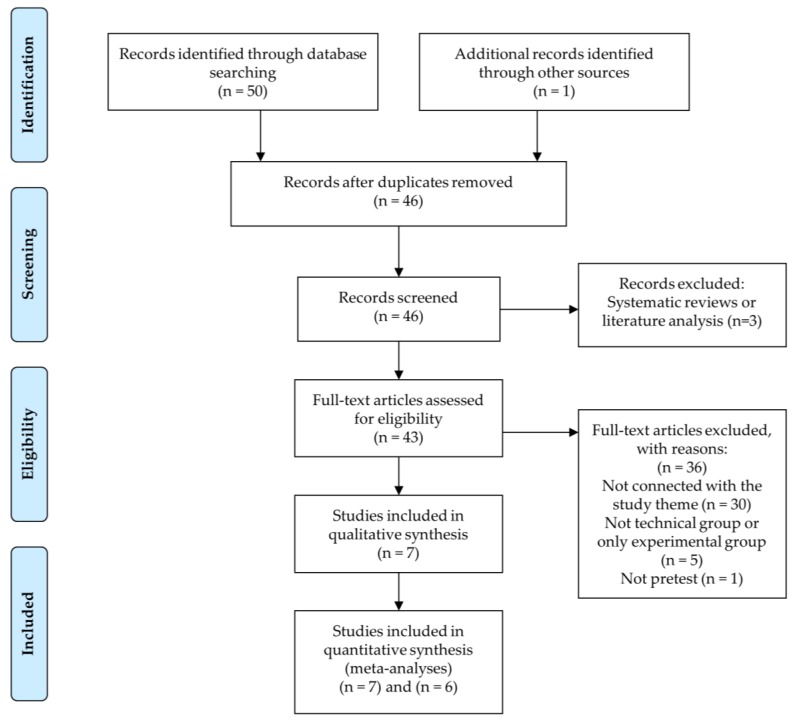
Flow diagram for the systematic review process according to Preferred Reporting Items for Systematic Review and Meta-Analysis Protocols (PRISMA) statements.

**Figure 2 ijerph-17-00505-f002:**
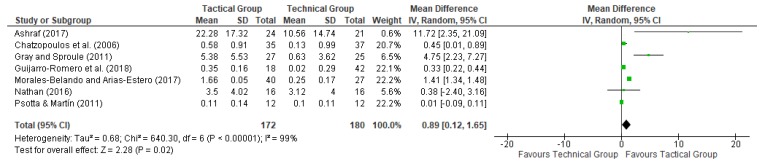
Meta-analysis results of the effects of TEG and TAG on decision making.

**Figure 3 ijerph-17-00505-f003:**
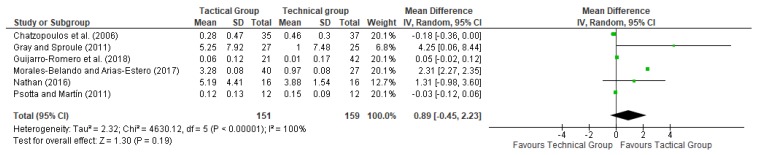
Meta-analysis results of the effects of TEG and TAG on skill execution.

**Table 1 ijerph-17-00505-t001:** Risk of bias according to the PEDro Scale.

	Response to Each Item Level of Evidence											
Study	1	2	3	4	5	6	7	8	9	10	11	Total Score
Guijarro-Romero et al., 2018	Y	N	Y	Y	Y	N	N	Y	Y	Y	Y	7
Ashraf 2017	N	Y	Y	Y	Y	N	N	Y	Y	Y	Y	8
Morales-Belando and Arias-Estero 2017	Y	Y	Y	Y	Y	N	N	Y	Y	Y	Y	8
Nathan 2016	N	Y	Y	Y	Y	N	N	Y	Y	Y	Y	8
Gray and Sproule 2011	N	N	N	Y	Y	N	N	Y	Y	Y	Y	6
Psotta and Martin 2011	Y	N	N	Y	Y	N	N	Y	Y	Y	Y	6
Chatzopoulos et al., 2006	N	Y	Y	Y	Y	N	N	Y	Y	Y	Y	8

Y: criterion fulfilled; N: criterion not fulfilled; 1: eligibility criteria were defined; 2: the participants were randomly distributed to groups; 3: the assignment was concealed; 4: the groups were similar before the intervention (at baseline); 5: all participants were blinded; 6: therapists (teachers) who conducted the intervention were blinded; 7: there was blinding of all evaluators; 8: the measures of at least one of the fundamental outcomes were attained from more than 85% of the participants initially; 9: “intention to treat” analysis was conducted for all participants who received the control condition or treatment as assigned; 10: the findings of statistical comparisons between groups were reported for at least one fundamental outcome; 11: the study gives variability and punctual measures for at least one fundamental outcome; total score: each satisfied item (except the first) adds 1 point to the total score.

**Table 2 ijerph-17-00505-t002:** Characteristics of the participants and the protocol.

Characteristics of the Sample				Protocol	
Study	Country	Sample Size of Groups and Sex	Age (SD) and Education Level/Setting	Tactical Group Treatment	Technical Group Treatment
Guijarro-Romero et al., 2018	Spain	TEG *: 42 (16 males and 26 females) TAGLIL ***: 23 (7 males and 16 female) In meta-analyses, it only used data from this group TAGHIL: 20 (16 males and 4 females)	10–12 years Primary school	Tactical approach	Technical approach
Ashraf 2017	Romania	TEG: 21 (NR ****) TAG **: 24 (NR)	20 (1.2) 20 (1.9) College students	Teaching Games for Understanding (TGfU)	Traditional method
Morales-Belando and Arias-Estero 2017	South of Europe	TEG: 27 (NR) TAG: 40 (NR) 45 males and 22 females (global data)	9.32 (2.60) (global data) Sailing school	Teaching Races for Understanding (TRfU)	Traditional teaching mode
Nathan 2016	Malaysia	TEG: 16 (8 females and 8 males) TAG: 16 (8 females and 8 males)	15.50 (1.00) (global data) Badminton school	TGfU revised	Skill Drill Technical
Gray and Sproule 2011	Scotland	TEG: 25 (12 females and 13 males) TAG: 27 (11 females and 16 males) * In meta-analyses, it used data on-the-ball “good” for decision making and data “successful” for skill execution	12.50 (0.20) 12.50 (0.30) Secondary school	Game-based approach	Skill-focused approach
Psotta and Martin 2011	Czech Republic	TEG: 12 (females) TAG: 12 (females)	21.00 (0.70) 20.70 (0.80) College students	Technical-tactical model with an emphasis on orientation to tactical	Technical-tactical model with an emphasis on orientation to technical skills
Chatzopoulos et al., (2006)	Greece	TEG: 37 (females) TAG: 35 (females)	12–13 years Middle school	Games approach	Technique approach

* TEG: Technical group; ** TAG: tactical group; *** TAGLIL: tactical group with a low initial tactical level; TAGHIL: tactical group with a high initial tactical level; **** NR: not reported.

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
