# Peer review of "Effects of Teaching Games on Decision Making and Skill Execution: A Systematic Review and Meta-Analysis"

_ijerph, 2020, doi:10.3390/ijerph17020505_

Round 1

Reviewer 1 Report

GENERAL COMMENTS:

The area is one of relevance and interest in the applied domain. Paper sufficiently advances knowledge to merit publication but I would like to highlight the following general limitations of the study:

The major limitation is the same as that of previous studies on this research topic that the authors highlighted, the lack of analysis on the influence of the teacher/coach management style. It is well known that the teaching outcomes not only depend on the teaching approach but at least at the same level as the coach's personality, her communication skills, the use of feedback in the sense of quantity and quality, motivation, etc. Given that the authors are analyzing the previous literature and to change that limitation is absolutely impossible, one of their contributions should be to include a deeper analysis of the description of these mediating factors in the documents included in the sample. In this sense, and in order to know if the previous investigation reported some type of information about these important mediators, it should be necessary to include at least one new file in one of the current tables.

According to the importance of this topic, after analyzing to what extent the documents include that information, both the introduction and the discussion should adequately consider the results and, just in case, the criticism of the lack of information about the coaches' teaching style, including this analysis as part of the objectives of the study.

ABSTRACT

Abstract reflects the work done and the conclusions drawn

P1; L28: After “…data base…” a full stop or a connector word should be convenient.

Following the previous recommendation, the new aim should be included in the abstract.

INTRODUCTION

The introduction of the article outlines the problem clearly and precisely, including the state of the question and raising the contributions of the study that would cover some of the shortcomings analyzed in previous research but following the highest limitation previously considered, criticism about the lack of information about coaches’ mediators must be included.

METHOD

In the same vein, the analysis of information that the papers of the sample include about coaches’ management must be done.

DISCUSSION

After analyzing the new results about coaches’ management style these must be accordingly discussed.

SOME MINOR RECOMMENDATIONS IN WRITING

P2, L59. The authors wrote:  “....develops”. The correct spelling is “develop" (without “s”) because it is referred to “tactical approaches”.

P4, L129. The authors wrote: “...evaluators; 8:the measures...”. A space between “8: and “...the measures” is needed.

P4, L130. The authors wrote: “...initially; 9:” intention to treat” when the should have written: initially; 9: “intention to treat” (space after colon and orientation of quotation marks).

P4, L131. The authors wrote: “...the study give variability...” The correct spelling is “gives".

P4, L190. I recommend substituting “in according to” by “in accordance with” or” “according to”.

Author Response

COVER LETTER

Manuscript ID: ijerph-687499. Type of manuscript: Article. Title: Effects of teaching games on decision making and skill execution: A Systematic review and meta-analysis

Reviewer 1’s comments and suggestions for authors

Details of the revisions and responses

We have removed table 3 at the request of reviewer 4.

We have included an analysis of the studies on these mediating factors (method and results)

According to the importance of this topic, after analyzing to what extent the documents include that information, both the introduction and the discussion should adequately consider the results and, just in case, the criticism of the lack of information about the coaches' teaching style, including this analysis as part of the objectives of the study.

We have included an analysis of the studies on these mediating factors (aim, introduction, discussion and conclusions).

Abstract reflects the work done and the conclusions drawn

P1; L28: After “…data base…” a full stop or a connector word should be convenient.

A full stop has been introduced.

SOME MINOR RECOMMENDATIONS IN WRITING

P2, L59. The authors wrote: “....develops”. The correct spelling is “develop" (without “s”) because it is referred to “tactical approaches”.

P4, L129. The authors wrote: “...evaluators; 8:the measures...”. A space between “8: and “...the measures” is needed.

P4, L130. The authors wrote: “...initially; 9:” intention to treat” when the should have written: initially; 9: “intention to treat” (space after colon and orientation of quotation marks).

P4, L131. The authors wrote: “...the study give variability...” The correct spelling is “gives".

P4, L190. I recommend substituting “in according to” by “in accordance with” or” “according to”.

These Recommendations have been done.

THANK YOU for your comments and suggestions

Reviewer 2 Report

This systematic review and meta-analyses analyzed the effects of technical and tactical approaches interventions on skill execution and decision making. The results of meta-analysis showed that tactical interventions achieved significant improvements on decision making and skill execution compared to technical approaches, though the heterogeneity of the interventions was large and the quality of the evidence was low according to GRADE.

I believe that this study can add knowledge to the field of education. I have only one minor concern on the adequateness of citations. The authors put reference 1 in the abstract. In the text, reference 2 appeared before reference 1. Reference 14 is missing……The authors need to examine the references and correct them.

Author Response

COVER LETTER

Manuscript ID: ijerph-687499. Type of manuscript: Article. Title: Effects of teaching games on decision making and skill execution: A Systematic review and meta-analysis

Reviewer 2’s comments and suggestions for authors

Details of the revisions and responses

The authors put reference 1 in the abstract. In the text, reference 2 appeared before reference 1.

We have corrected this references

Reference 14 is missing……The authors need to examine the references and correct them.

We have examined the references and corrected them

THANK YOU for your comments and suggestions

Reviewer 3 Report

Authors performed a meta-analysis review on the effects of teaching games on decision making and skill execution. The applied methodology is coherent with the common approach to both systematic and meta review. The review is also well-written and the results clearly presented. 

My only suggestion is to write in the main text the contents of Table 3 to increase the readability and, consequenlty to remove the Table 3. 

Author Response

COVER LETTER

Manuscript ID: ijerph-687499. Type of manuscript: Article. Title: Effects of teaching games on decision making and skill execution: A Systematic review and meta-analysis

Reviewer 3’s comments and suggestions for authors

Details of the revisions and responses

My only suggestion is to write in the main text the contents of Table 3 to increase the readability and, consequently to remove the Table 3. 

We have considered this suggestion.

THANK YOU for your comments and suggestions

Reviewer 4 Report

abstract. Please define better the assessed outcomes into the aims abstract  regarding the % Confidence Interval (CI) from 0.65 to 2.92 etc.. please define the unit of measures in brackets  abstract. the conclusion is not in line with the aim of this study. change in according to the results page 6. add one column regarding the (country) and if any disease table 4 and 5 should be as appendix. the same results are showed as a forest plot.  regarding the forest plot please define them in a better resolution meta-easy is a very elementary software for conducting meta analysis, please revise EXTENSIVELY ALL the analysis using the software review manager 5 (download it, it is free). It is a better software that all You to show into the forest plot directly the analysis on random effects and the etherogenity test..ans so on..  

Author Response

COVER LETTER

Manuscript ID: ijerph-687499. Type of manuscript: Article. Title: Effects of teaching games on decision making and skill execution: A Systematic review and meta-analysis

Reviewer 4’s comments and suggestions for authors

Details of the revisions and responses

Abstract. Please define better the assessed outcomes into the aims regarding the % Confidence Interval (CI) from 0.65 to 2.92 etc. Please define the unit of measures in brackets.

Effect size and CI have no units. We have defined the assessed outcomes in the abstract.

The conclusion is not in line with the aim of this study.

We have related conclusion and aim.

add one column regarding the (country).

We have added one column (Table 2) regarding the country.

Tables 4 and 5 should be as appendix.

We have included that information in the Figures from RevMan5.

The same results are showed as a forest plot. regarding the forest plot please define them in a better resolution meta-easy is a very elementary software for conducting meta analysis, please revise EXTENSIVELY ALL the analysis using the software review manager 5 (download it, it is free). It is a better software that all You to show into the forest plot directly the analysis on random effects and the etherogenity test..ans so on..

We have checked the analyses using RevMan 5 and made the consequent changes.

THANK YOU for your comments and suggestions

Round 2

Reviewer 4 Report

all my comments during the first revision have been addressed.